# Does Insomnia Increase the Risk of Suicide in Hospitalized Patients with Major Depressive Disorder? A Nationwide Inpatient Analysis from 2006 to 2015

**DOI:** 10.3390/bs12050117

**Published:** 2022-04-19

**Authors:** Zeeshan Mansuri, Abhishek Reddy, Ramu Vadukapuram, Mounica Thootkur, Chintan Trivedi

**Affiliations:** 1Department of Psychiatry, Boston Children’s Hospital, Harvard Medical School, Boston, MA 02115, USA; 2Department of Psychiatry, Virginia Tech Carilion School of Medicine, Roanoke, VA 24016, USA; mrthootkur@carilionclinic.org; 3Department of Psychiatry, Icahn School of Medicine at Mount Sinai, New York, NY 10029, USA; ramu.vadukapuram@mssm.edu; 4Department of Psychiatry, Texas Tech University Health Science Center at Permian Basin, Odessa, TX 79763, USA; chintan.trivedi@ttuhsc.edu

**Keywords:** insomnia, depression, suicide

## Abstract

**Introduction.** Insomnia is an important symptom associated with major depressive disorder (MDD). In addition, it is one of the risk factors for suicide. Studies have shown the relationship be-tween insomnia and suicidal behavior in patients with MDD. However, this association has not been evaluated in a large sample of hospitalized patients. **Objectives.** To evaluate the suicidal be-havior in MDD patients with insomnia compared to those without insomnia. **Methods.** From the National Inpatient Sample (NIS 2006–2015) database using the ICD-9 code, patients’ data were obtained with the primary diagnosis of MDD and comorbid diagnosis of insomnia disorders (MDD+I). These patients were compared with MDD patients without insomnia disorders (MDD–I) by performing a 1:2 match for the primary diagnosis code. Suicidal ideation/attempt da-ta were compared between the groups by multivariate logistic regression analysis. **Results.** After the diagnostic code matching, 139061 patients were included in the MDD+I group and 276496 patients in the MDD–I group. MDD+I patients were older (47 years vs. 45 years, *p* < 0.001) com-pared to the MDD–I group. The rate of suicidal ideation/attempt was 56.0% in the MDD+I group and 42.0% in the MDD–I group (*p* < 0.001). After adjusting for age, sex, race, borderline personal-ity disorders, anxiety disorders, and substance use disorders, ‘insomnia’ was associated with 1.71 times higher odds of suicidal behavior among MDD patients admitted to the hospital. (Odds ratio: 1.71, 95% confidence interval 1.60–1.82, *p* < 0.001). Conclusions. Insomnia among MDD patients is significantly associated with the risk of suicide. MDD patients with insomnia need to be closely monitored for suicidal behavior.

## 1. Introduction

Growing consideration has been given to suicide worldwide, with several countries implementing national strategies for prevention [1]. According to the CDC (Centers for Disease Control and Prevention), in the year 2018, 10.7 million adults seriously thought about suicide, 3.3 million made a plan, and 1.4 million attempted suicide in the United States [2]. In 2017, 1.4% deaths occurred because of suicide all over the world [3]. The 10th leading cause of death in the United States among all the ages is suicide. Suicide is a chief contributor to early death as it is the second leading cause of death in ages 25–34 and the third leading cause in ages 10–24 in the United States [4]. According to National Center for Health Statistics (NCHS) data from 1999 to 2018, the age-adjusted suicide rate raised by 35% with males having 3.5 to 4.5 times higher suicide rates compared to females over the entire period (1999–2018) [4]. Higher suicide rates are concerning among individuals with chronic psychiatric illnesses, including substance abuse disorders, and in individuals with a history of previous suicide attempts [5,6]. Out of all psychiatric illnesses, the rate of suicide in individuals with major depressive disorder (MDD) is especially concerning. According to WHO reports, around 2% to 15% of patients with MDD end their life by suicide [7]. MDD is associated with increased functional disability and mortality. In 2017, around 11 million individuals aged 18 or older had at least one MDD episode with severe impairment, representing 4.5% of all U.S. adults [8]. The incremental economic burden of individuals with MDD in the U.S. increased by 21.5%, exceeding 200 billion USD in the year 2010 [9]. Several studies have explored the factors responsible for suicide in individuals with MDD and suggested that alcohol abuse, a low level of social and occupational functioning, and poor perceived social support [10] can be potential risk factors for suicide. In addition to these risk factors, sleep disturbances may constitute as one of the modifiable risk factors for suicidal behaviors [11,12]. Sleep problems are defining features of several psychiatric disorders and are included among the diagnostic criteria for many of these disorders, including depression [13]. Disturbed sleep is a very distressing symptom which has a huge impact on the quality of life in patients with MDD [14]. The majority of patients with MDD present with insomnia symptoms [15].

Insomnia is a common sleep disorder, characterized by difficulty falling or staying asleep, and it is associated with significant distress or impairment during the daytime and happens despite sufficient opportunity for sleep [16]. Insomnia and depression often present together. Though up to 24% of people with depression present with excessive sleep [17], as many as 84.7% have trouble falling asleep or staying asleep [18]. Cross-sectional studies have shown a strong association between symptoms of depression and insomnia, and insomnia is longitudinally associated with the development of depression and poor treatment outcomes [19,20]. There exists a bidirectional relationship between insomnia and mood disorders, according to studies. In patients with insomnia and mood disorders, insomnia preceded the mood disorder in 41% of cases, the mood disorder preceded insomnia in 29% of cases, and the symptoms appeared at the same time in 29% of cases [21]. Insomnia is reported as an independent modifiable risk factor for suicide attempts [22,23] and suicide [24,25]. Insomnia, in particular, may confer higher risk for suicidal behavior in individuals with depression [24] and insomnia’s association with suicidal ideation may be better explained by the presence of depression [26]. The mechanism behind sleep disturbances increasing the risk of suicidal behavior remains unclear. However, it is proposed that sleep disturbances worsen psychological suffering, rendering psychiatric patients more vulnerable to suicidal behavior to reduce or escape from such distress [11,12,27]. In addition to that, fatigue after sleep difficulties may impair problem solving and decrease emotion regulation, increasing one’s risk for suicidal behavior through vulnerability to impulsive behavior [28]. Insomnia may be considered a clinical indicator of acute suicidal risk, particularly when it appears during a depressive episode [29]. Noting the observable and modifiable risk factors of suicidal behavior and understanding the relationship between insomnia and suicide in individuals with MDD is critical for suicide prevention efforts and providing high-quality care to our patients. There is a dearth of literature observing the relationship of MDD, insomnia, and suicide in the inpatient population. Thus, with this study, our main objective is to evaluate suicidal behavior in MDD patients with insomnia compared to those without insomnia from a nationwide inpatient sample from the years 2006–2015.

## 2. Methods

### 2.1. Data Source

Study data from January 2006 to October 2015 were obtained using the Nationwide Inpatient Sample (NIS) dataset [30]. The NIS dataset is developed for the Healthcare Cost and Utilization Project (HCUP) sponsored by the Agency for Healthcare Research and Quality (AHRQ). The dataset is released every year and contains information on more than 200 patient- and hospital-level variables. For each record, the discharge level weight is provided to calculate national estimates. It is the largest publicly available inpatient healthcare database designed to produce U.S. regional and national estimates of inpatient utilization, access, charges, quality, and outcomes, irrespective of insurance status.

Patient-level characteristics in the dataset are age, gender, and race. The hospital-level characteristics include hospital region, length of stay, total charge, and discharge disposition. Also, it is possible to estimate total cost from the cost to charge ratio (CCR) files provided by the NIS dataset. Diagnostic codes [International Classification of Diseases (ICD), nine and ten versions] for disease and procedures are provided in the dataset. By querying the specific diagnostic code, data on specific diagnoses can be collected.

### 2.2. Patient Population

#### 2.2.1. Study Group

The primary population was composed of patients with a primary diagnosis of major depressive disorder and a secondary diagnosis of insomnia (MDD+I group). The ICD-9 code used to obtain our target population is provided in Appendix A.

#### 2.2.2. Control Group

The control population was composed of MDD patients without insomnia diagnosis (MDD−I group). Data on age, gender, race, substance use disorders, discharge disposition, length of stay, total hospitalization cost, substance use disorders, and other psychiatric disorders were collected. Data on psychiatric comorbidities were collected based on the NIS dataset guidelines [31].

#### 2.2.3. Outcome

The primary outcome was a composite outcome of suicidal ideation/attempt. ICD-9 code V6284 was used to collect suicidal ideation data, and E950-E959 was used to collected suicide attempt data from the dataset.

#### 2.2.4. Covariates

Covariates included were age, gender, race, alcohol abuse, anxiety disorders, substance use disorders, and borderline personality disorders.

### 2.3. Statistical Analysis

#### 2.3.1. Descriptive Statistics and Propensity Matching

Descriptive statistics were performed for the baseline characteristics and presented in a table format. Mean (standard error) was used to present continuous data, and the percentage was used for categorical data. To balance the burden of depression between the groups, control group was selected by performing a 1:2 match with the primary population. The nearest neighbor propensity score matching technique was used from random order using a caliper size of 0.001. Propensity scores were computed and matched. Matching was performed for the primary diagnosis code of MDD (covariate), and insomnia (MDD+I (insomnia: yes) vs. MDD−I (insomnia: no)) was used as an outcome in the propensity score-matched analysis.

#### 2.3.2. Logistic Regression Analysis

The association of insomnia diagnosis to suicidality in MDD patients was analyzed using logistic regression methods. In Model 1, age, gender, race, alcohol abuse, substance use disorders, anxiety disorders, borderline personality disorders, and study group (MDD+I vs. MDD−I) were included as covariates. Model 2 was performed only for the MDD+I group for the same covariates. Model 3 was performed for same variables included in Model 1, but only suicide ideation was included as an outcome.

All tests were two-sided, and a *p*-value of less than 0.05 was considered statistically significant. The statistical analysis was performed using SPSS version 26.0 software for Windows (IBM Software, Inc, Armonk, NY, USA) by complex sample analysis procedures. In addition, R Foundation for statistical computing version 3.6.3. was used to perform propensity score matching.

## 3. Results

From the dataset, 139,061 patients with the primary diagnosis of MDD and secondary diagnosis of insomnia were included as a primary population. The control group was selected by performing a 1:2 match for the diagnostic code of MDD to balance the burden of depression (Appendix A). After matching, 139,061 patients were included in the MDD+I group, and 276,496 patients were included in the MDD−I group.

### 3.1. Baseline Characteristics

Baseline patient and hospital characteristics are shown in Table 1. MDD+I patients were older (47 years vs. 45 years, *p* < 0.001) compared to the MDD−I group. More patients were female (58.7% vs. 57.4%) and were white (76.7% vs. 74.7%) in both groups. The prevalence of adjustment disorders, personality disorders, alcohol abuse, and substance use disorder was similar between the groups. There were more anxiety and psychotic disorders in the MDD+I group. There was more obstructive sleep apnea in the MDD+I group, and there was more hypothyroidism in the MDD−I group.

### 3.2. Suicidal Behaviors

The rate of suicidal ideation and suicidal attempt is shown in Figure 1. More patients in the MDD+I group had suicidal ideation (51.7% vs. 38.0%, *p* < 0.001). There was no difference in the incidence rate of suicidal attempt (5.6% vs. 5.5%, *p*: 0.47). The composite outcome of suicidal ideation/attempt was 55.5% and 42.4% in the MDD+I and MDD−I groups, respectively (*p* < 0.001).

In the logistic regression analysis (Model 1, Table 2), after adjusting for age, gender, race, alcohol abuse, substance use disorders, and borderline personality disorders, MDD+I was associated with 71% more risk of suicidality (suicidal ideation/attempt) (Odds ratio: 1.71, *p* < 0.001) compared to MDD−I.

Anxiety disorders (OR: 1.27, *p* < 0.001), alcohol abuse (OR: 1.20, *p* < 0.001), substance abuse (OR: 1.14, *p* < 0.001), and borderline personality disorders (OR: 1.39, *p* < 0.001) were other covariates which showed an association with suicidality. Suicidality was less in female (OR: 0.82, *p* < 0.001) and older patients (increase of 5 years in age, OR: 0.92, *p* < 0.001).

In the separate model (Model 2) for the MDD+I group, borderline personality disorders were associated with 38% more odds of suicidality and anxiety disorders were associated with 23% higher odds of suicidality. There was also an association of alcohol and substance abuse with a higher rate of suicidality. Odds of suicidality were 20% less in females (compared to male) and 9% less with a 5-year increase in age. When only suicide ideation was included as an outcome (Model 3), odds of association between insomnia and suicidal ideation were 1.71 times compared to those without insomnia.

## 4. Discussion

To the best of our knowledge, our study is unique because we used a large sample such as the Nationwide Inpatient Sample (NIS data 2006–2015) to analyze the relation between MDD, insomnia, and suicide and found a strong association. We found a significantly higher suicidal ideation/attempt in MDD patients with insomnia (55.5%) when compared to MDD patients without insomnia (42.4%) (*p* < 0.001). Even after adjusting for confounding factors, insomnia was associated with 71% more risk of suicidality compared to MDD patients without insomnia. Similar results were seen in a retrospective study by Agargun et al. They noticed patients with MDD having insomnia and/or hypersomnia were significantly more likely (*p* < 0.001) to become suicidal than the patients in the no sleep disturbance group. Likewise, Gallagher et al. reported data on 163,512 depressed patients with a history of sleep disturbance, where 1.0% of patients had suicidal ideation compared to 0.7% of patients with no history of sleep disturbance, whereas 0.3% patients with a history of sleep disturbance had attempted suicide in comparison to 0.2% in patients without a history of sleep disturbance [32].

In our study, patients in the MDD+I group had higher suicidal ideation (51.7% vs. 38.0%, *p* < 0.001) in comparison to the MDD−I group. Similar association was observed in a cross-sectional study by Chellappa, S. L. et al.

In our study, patients in the MDD+I group were older and had a high prevalence of anxiety disorders and psychotic disorders in comparison to the MDD−I group. Other studies also show similar results. [33,34].

In our study, patients in the MDD+I group had a longer length of stay and a higher total hospital cost compared to MDD−I. Likewise, a retrospective, observational study compared direct health care costs in patients with MDD+I and MDD−I in adults. The direct costs for the MDD+I group were significantly (*p* < 0.001) higher than MDD−I (USD 4858 vs. USD 4007). Similar direct cost differences (USD 1007) were found in a sample of the elderly in a study by Asche et al. [35].

Several hypotheses have been proposed to understand the link between insomnia and suicide in MDD patients. The role of serotonin has been identified in the role of insomnia and its effect on suicidal behavior in depressed patients [36,37]. Dysfunctional beliefs and attitudes about sleep play an important role in insomnia, MDD, and suicide risk [38]. Hopelessness, a common symptom in MDD, plays a role in dysfunctional belief leading to chronic insomnia and is potentially a risk factor for suicide. Increased nocturnal wakefulness can also result in increased suicidal tendencies in patients with depression [38,39].

In summary, our study shows a strong association between suicidal ideation/attempt in MDD patients with insomnia as compared to patients without insomnia in the hospitalized patients dataset. Our study also notes that patients with MDD and insomnia were in the older age group and had higher prevalence of anxiety and psychotic disorders. Also highlighted in our analysis is the fact that hospitalized patients with MDD and insomnia had a longer length of stay and the costs of hospitalization were higher. Thus, we highlight the importance of monitoring and treating sleep disturbances in patients admitted to hospitals to prevent worsening mental health and also to reduce costs of hospitalizations.

Further studies are needed in exploring the observable and modifiable risk factors of suicidal behavior and understanding the relationship between insomnia and suicide in individuals with MDD as this could be crucial for suicide prevention efforts and providing high-quality care to our patients.

## 5. Limitations

One of the major limitations of the study is the observational study design, which is prone to selection and confounding bias. The ICD coding system provided by the NIS dataset is prone to coding errors and the under-reporting of comorbidities. It is not possible to validate the ICD code because of the nature of the dataset. The NIS database is limited to inpatient data only, so it is not possible to collect longitudinal data or outpatient data. The NIS database records are of individual admissions, not patients, so there is a chance that patients can be counted more than once. Despite these limitations, it is counterbalanced by a significantly large sample size and propensity score matching technique, which could have reduced the selection bias.

## 6. Conclusions

There is an increased risk of suicide in MDD patients with insomnia. Clinicians need to be watchful for sleep disturbances in this patient population and should address it in a timely manner. Our findings underline the importance of screening for insomnia in MDD patients to prevent the risk of suicidal behavior. Further, large-scale randomized studies are needed to explore this association. Studies should also focus on specific sleep disturbances and their effect on the risk of suicidality in a differential manner.

## Figures and Tables

**Figure 1 behavsci-12-00117-f001:**
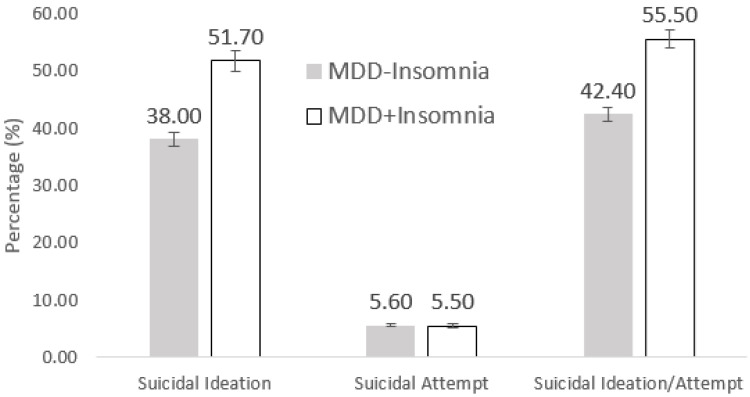
Rate of suicidality between two groups.

**Table 1 behavsci-12-00117-t001:** Baseline Patient and Hospital Characteristics of study population.

	MDD without Insomnia (N = 276,496)	MDD with Insomnia (N = 139,061)	*p*-Value
Age			
Mean, SE	45.1 (0.14)	47.4 (0.20)	<0.001
Sex			0.002
Male	42.6	41.3	
Female	57.4	58.7	
Race			<0.001
White	74.70	76.7	
Black	12.60	11.30	
Hispanic	7.90	7.30	
Asian or Pacific Islander	1.20	1.30	
Native American	0.70	0.50	
Other/unknown	2.90	2.90	
Comorbidities			
Personality disorders	14.70	13.70	0.06
Anxiety disorders	31.90	53.90	<0.001
Adjustment disorders	1.90	2.10	0.22
Psychotic disorders	3.20	4.20	<0.001
Alcohol abuse	24.90	24.10	0.11
Substance abuse	28.20	28.60	0.51
Obstructive sleep apnea	3.2	3.9	<0.001
Hypothyroidism	10.7	8.9	<0.001
Hospital region			0.02
Urban	86.70	88.70	
Rural	13.30	11.30	
Disposition			<0.001
Routine	87.70	88.40	
Other health care facility	10.80	10.50	
Against medical advice (AMA)	1.50	1.00	
Length of hospital stay, days	6.58 (0.07)	7.45 (0.09)	<0.001
Total cost of hospitalization, USD	5077 (73)	6026 (86)	<0.001

MDD+I patients had a longer length of stay, and the total hospitalization cost was more. However, more patients in the MDD−I group were discharged to other healthcare facilities or against medical advice.

**Table 2 behavsci-12-00117-t002:** Predictors associated with suicidal ideation/attempt among study population.

	Model 1		Model 2		Model 3	
	Odds Ratio (95% Confidence Interval)	*p*-Value	Odds Ratio (95% Confidence Interval)	*p*-Value	Odds Ratio (95% Confidence Interval)	*p*-Value
Age, per 5 years increase	0.923 (0.917–0.928)	<0.001	0.912 (0.903–0.921)	<0.001	0.939 (0.934–0.945)	<0.001
Female	0.817 (0.791–0.843)	<0.001	0.795 (0.755–0.837)	<0.001	0.777 (0.753–0.802)	<0.001
Non-white Race	1.031 (0.968–1.099)	0.43	0.996 (0.914–1.086)	0.43	1.062 (0.996–1.132)	0.14
Alcohol abuse	1.201 (1.157–1.247)	<0.001	1.201 (1.126–1.281)	<0.001	1.156 (1.106–1.195)	<0.001
Substance use disorders	1.137 (1.087–1.188)	<0.001	1.071 (0.998–1.149)	<0.001	1.161 (1.12–1.212)	<0.001
Borderline personality disorders	1.387 (1.290–1.492)	<0.001	1.377 (1.215–1.561)	<0.001	1.274 (1.183–1.372)	<0.001
Anxiety disorders	1.273 (1.208–1.342)	<0.001	1.233 (1.124–1.352)	<0.001	1.295 (1.227–1.367)	<0.001
MDD + insomnia	1.705 (1.596–1.822)	<0.001	--	--	1.708 (1.615–1.848)	<0.001

In Model 1, age, gender, race, alcohol abuse, substance use disorders, anxiety disorders, borderline personality disorders, and study group (MDD+I vs. MDD−I) were included as covariates and suicidal ideation/attempt was included as an outcome. Model 2 was performed only for the MDD+I group for the same covariates with the same outcome. Model 3 was performed for same variables included in Model 1, but only suicide ideation was included as an outcome.

## Data Availability

Dataset is available at https://www.hcup-us.ahrq.gov/, accessed on 7 January 2020.

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
