# Peer review of "Does Insomnia Increase the Risk of Suicide in Hospitalized Patients with Major Depressive Disorder? A Nationwide Inpatient Analysis from 2006 to 2015"

_behavsci, 2022, doi:10.3390/bs12050117_

Round 1

Reviewer 1 Report

Thank you for inviting me to review this manuscript on the association between insomnia and suicidal ideation/attempt among inpatients with MDD. The study used a national inpatient sample dataset in the US with ICD code for MDD, insomnia, and other disorders, and found that the incidence of suicidal ideation/attempt was higher in MDD patients with insomnia. Despite the observational nature, the study highlighted the need to be watchful for sleep disturbances in MDD patients to prevent the risk of suicide.

Introduction:

  • What does CDC stand for?

Methods

  • In statistical analysis, it says “To balance the burden of depression between the groups' control group was selected by performing a 1:2 match with the primary population”. Would the authors explain more about the rationale of doing a propensity score matching on the primary diagnosis code of MDD please?

Results

  • Are “substance abuse” in Table 1 and “substance use disorders” in the logistic regression referring to the same condition?
  • Why were borderline personality disorder and anxiety disorder, but not other personality disorders and psychiatric disorders, chosen to be included in the logistic regression?

Discussion

  • In the second paragraph, it says “(3.68+/-1.73) [36]. In our study we didn’t see a statistically significant difference in the incidence rate of suicidal attempt (5.6% vs. 5.5%, p: 0.47), whereas in this cross-sectional study, depressed patients with insomnia had significantly higher scores on the components such as active suicidal ideation, specific plans for suicide, and previous suicide attempts [37].” Are there any potential explanations for the different results between the studies?
  • In the third paragraph, it says “These participants were followed for one year and found that twelve participants developed MDD. This study suggested that elderly participants with persistent insomnia are at greater risk for the development of new onset depression.” It reads like the twelve participants were all with persistent insomnia when this is not the case in that study.
  • In the third paragraph, what does SUD stand for?

Author Response

Attached file.

Reviewer 2 Report

Thank you for the opportunity to review this interesting paper on the link between insomnia disorder and suicidal behaviours in individuals with a diagnosis of major depressive disorder. In general, I found the paper to be interesting and relevant to the journal. I have highlighted my concerns about the paper which means I do not think the manuscript in its current format is ready for publication. I have listed my concerns below starting with the more major issues and following on with minor issues relevant to each relevant section.

My first concern relates to the outcome measure which is a composite measure combining a question on suicidal ideation and a question on suicidal behaviour. Whilst I have no issue with the questions being used, the combination of suicidal thinking and behaviour into one measure is somewhat at odds with major thinking in the area of suicidology where focus is very much shifting to an ideation to action approach to suicide. With this approach, suicidal ideation and behaviour would be considered separately rather than jointly on the basis that those variables which may predict suicidal ideation, do not necessarily also predict behaviour. In this study, the results suggests differences between the two groups on suicidal ideation, but not behaviour, which again to my mind would lend itself to examining these separately rather than as a composite. Thus, I think a justification for considering these together would be helpful and some consideration of the results in relation to the theoretical work in suicidology would be useful.

Following on from this, I wonder if the authors could clarify the choice of covariates and also why some, e.g., alcohol abuse and substance abuse were included as covariates when there were no differences between the two groups on these variables. In addition, when looking at the aim of the study “Our main objective is to evaluate suicidal behavior in MDD patients with insomnia compared to those without insomnia from nationwide inpatient sample from the year 2006-2015” it is not clear how all of the presented analyses really help to address that issue. For example, the model 2 analysis does not seem relevant. It would therefore be helpful with some clearer linkages between aims and analyses.

The discussion doesn’t really make attempts at understanding all of the findings. For example, the finding that the two groups did not differ on suicidal behaviour was mentioned but not discussed. Going back to my previous point on ideation to action frameworks, I think there is a way to make sense of this finding within such a framework. Overall, I found that the discussion really was more focussed on previous research rather than making sense of the findings within the current study.

Finally, I felt the manuscript would benefit from being proof read as there are a number of awkward sentences and topographical errors.

Intro:

  • In the opening paragraphs, reference is made to both international rates and US rates; however, at times it is not clear which is it. for example, in the sentence which starts “suicide is a chief contributor to early death..” it is not clear if this is for the US or the world.

Methodology:

  • In table 1 I am unsure about what ‘against medical advice’ refers to. It would be helpful to clarify this.

Table S2: I did not find this table to be particularly helpful and wonder if it could be removed

Author Response

attached file.

Reviewer 3 Report

In order to examine the potential relationship between insomnia and suicide risk in hospitalized MDD patients, the authors investigated the role of insomnia on suicidal ideation/attempt in MDD patients. For this aim, they identified two groups, MDD patients with insomnia and MDD patients without insomnia, respectively.

Secondarily to the Multivariate logistic regression analysis, the authors found that the MDD+I group was associated with more risk of suicidal behavior with respect to the MDD-I group. The results demonstrated that in MDD the comorbidity of sleep disorder represents a risk factor for suicidal ideation/attempt.

In my opinion, the present paper is interesting and well written. The methodological and statistical procedure is correct, although they should be explained better and conclusions are logically linked to the results. I feel that it is a relevant paper for readers of Behavioral Sciences.

I do have some concerns:

-Introduction: 

In the introduction, a mention could be made on the direction of the relationship between insomnia and mood disorders, considering the close bidirectional relationship that exists between the two clinical conditions and the difficulty of detecting the direction of this relationship.

-Discussion:

The results could be discussed more in deep according to recent articles that can support authors both in the discussion and in the introduction. Furthermore, the authors could better describe the psychological and behavioral alterations that prolonged hospitalization can cause and the impact it has on the quality and quantity of sleep, regardless of the diagnosis of insomnia.

Author Response

attached file.

Round 2

Reviewer 2 Report

Thank you to the authors for replying to my comments. Unfortunately some of my initial concerns still apply as although the analysis relating to suicidal ideation has now been included, the rationale for this still doesn't exist. To my mind, the introduction still requires some effort to place this within the context of ideation to action frameworks. It does not make sense to have an aim which refers to suicidal behaviors but then the analyses are broken down. I think it would substantially strengthen the paper to recognise the move within the suicidology literature towards ideation-to-action frameworks and the subsequent value in conducting analyses separately for behaviour and thoughts. Thus, at the moment, I still do not think that the aim aligns with the analyses.

Linked to this, and for clarity, I would recommend that table 2 is updated with notes to ensure that it is clear when looking at this table what models 1, 2 and 3 refers to. The title of the table simply refers to suicidality which is a broad term.

In my previous comments I had asked why such a focus was placed on substance misuse. The response simply states that it has been highlighted as a risk factor in the literature. Whilst I agree with this, there are a number of risk factors which have been highlighted as important, so I am still looking for some reasoning for focussing on this specific one over others.

There are also still a number of typographical issues which should be addressed to ensure the overall readability of the document is improved.
